# The Therapeutic Potential of Butyrate and Lauric Acid in Modulating Glial and Neuronal Activity in Alzheimer’s Disease

**DOI:** 10.3390/nu17142286

**Published:** 2025-07-10

**Authors:** Rathnayaka Mudiyanselage Uththara Sachinthanie Senarath, Lotta E. Oikari, Prashant Bharadwaj, Vijay Jayasena, Ralph N. Martins, Wanakulasuriya Mary Ann Dipika Binosha Fernando

**Affiliations:** 1School of Medical and Health Sciences, Edith Cowan University, Joondalup, WA 6027, Australia; rsenarat@our.ecu.edu.au (R.M.U.S.S.); p.bharadwaj@ecu.edu.au (P.B.); r.martins@ecu.edu.au (R.N.M.); 2Australian Alzheimer’s Research, Ralph and Patricia Sarich Neuroscience Research Institute, Nedlands, WA 6009, Australia; 3Brain and Mental Health, QIMR Berghofer Medical Research Institute, Brisbane, QLD 4006, Australia; lotta.oikari@qimrberghofer.edu.au; 4Nutrition and Food Science, School of Science, Western Sydney University, Campbelltown, NSW 2560, Australia; v.jayasena@westernsydney.edu.au; 5Lions Alzheimer’s Foundation, Nedlands, WA 6909, Australia; 6Department of Biomedical Sciences, Macquarie University, Sydney, NSW 2109, Australia

**Keywords:** Alzheimer’s disease, astrocytes, neurons, microglia, butyrate, lauric acid

## Abstract

Alzheimer’s disease (AD) is a progressive neurodegenerative disorder marked by amyloid-β plaque accumulation, tau tangles, and extensive neuroinflammation. Neuroinflammation, driven by glial cells like microglia and astrocytes, plays a critical role in AD progression. Initially, these cells provide protective functions, such as debris clearance and neurotrophic support. However, as AD progresses, chronic activation of these cells exacerbates inflammation, contributing to synaptic dysfunction, neuronal loss, and cognitive decline. Microglia release pro-inflammatory cytokines and reactive oxygen species (ROS), while astrocytes undergo reactive astrogliosis, further impairing neuronal health. This maladaptive response from glial cells significantly accelerates disease pathology. Current AD treatments primarily aim at symptomatic relief, with limited success in disease modification. While amyloid-targeting therapies like Aducanumab and Lecanemab show some promise, their efficacy remains limited. In this context, natural compounds have gained attention for their potential to modulate neuroinflammation and promote neuroprotection. Among these, butyrate and lauric acid are particularly notable. Butyrate, produced by a healthy gut microbiome, acts as a histone deacetylase (HDAC) inhibitor, reducing pro-inflammatory cytokines and supporting neuronal health. Lauric acid, on the other hand, enhances mitochondrial function, reduces oxidative stress, and modulates inflammatory pathways, thereby supporting glial and neuronal health. Both compounds have been shown to decrease amyloid-β deposition, reduce neuroinflammation, and promote neuroprotection in AD models. This review explores the mechanisms through which butyrate and lauric acid modulate glial and neuronal activity, highlighting their potential as therapeutic agents for mitigating neuroinflammation and slowing AD progression.

## 1. Introduction

Alzheimer’s disease (AD) is a progressive, irreversible degenerative disorder affecting the hippocampus, entorhinal, and temporal-parietal cortex [1]. Main neuropathological hallmarks are the accumulation of extracellular plaque of amyloid-β peptide (Aβ) and the neurofibrillary tangles (NFT) of the microtubule-binding tau protein [2]. Pathological conditions associated with AD trigger neuroinflammation, which includes a wide range of inflammatory events that affect the central nervous system (CNS). As AD progresses, it causes a range of clinical symptoms from mild spontaneous cognitive impairment to severe neurological and psychological symptoms at advanced stages [3]. The progression of AD is influenced by neuroinflammation and cellular dysfunction, both of which play critical roles in the pathophysiology of the disease [4]. Neuroinflammation refers to the inflammatory response within the CNS, primarily involving glial cells like microglia and astrocytes [5]. Glial cells are non-neuronal cells in the CNS that support, protect, and maintain neuronal function [6]. Microglia are the primary immune cells of the CNS and act as the first line of defense against injury and infection. They are crucial for synaptic pruning, debris clearance, and the release of neurotrophic factors [7]. Astrocytes, on the other hand, provide structural support, regulate the blood–brain barrier (BBB), and maintain homeostasis by managing neurotransmitter levels and ion balance [8].

As AD progresses, microglia and astrocytes can shift from protective to detrimental roles. Initially, microglia help clear amyloid plaques and reduce damage, while astrocytes provide neurotrophic support and maintain synaptic function [9,10]. However, chronic activation of these cells in response to AD pathology leads to a pro-inflammatory state [11]. Microglia release cytokines such as tumor necrosis factor (TNF)-α, interleukin (IL)-1β, and IL-6, along with reactive oxygen species (ROS), which damage synapses, promote neuronal apoptosis, and perpetuate neuroinflammation [12]. Similarly, astrocytes undergo reactive astrogliosis, a process initially aimed at protecting neurons. Prolonged activation, however, results in the loss of their supportive functions, exacerbating inflammation and further impairing cognitive functions [13]. This maladaptive response from microglia and astrocytes amplifies synaptic dysfunction, neuronal loss, and cognitive decline, making their chronic dysregulation a significant driver of AD progression [14]. Recent literature also emphasizes the importance of neuroimmune interactions mediated by microglial polarization and NLRP3 inflammasome activation as key drivers of AD progression [15].

Current treatments for AD primarily focus on symptomatic relief, with limited breakthroughs in disease modification. Aducanumab, the first FDA-approved disease-modifying drug, is being discontinued [16], while Lecanemab (Leqembi^®^), approved for early-stage AD, targets Aβ to slow cognitive decline [17]. Alongside these advancements, natural compounds are gaining attention for their potential to modulate neuroinflammation and promote neuroprotection [18]. Substances such as curcumin [19], phenols [20], and fatty acids [21] have shown promise in reducing pro-inflammatory pathways, protecting neurons, and resolving inflammation.

Among these, butyrate, a short-chain fatty acid (SCFA), and lauric acid, a medium-chain fatty acid (MCFA), are particularly notable. Butyrate, produced by a healthy gut microbiome, regulates the gut–brain axis, decreases neuroinflammation, and supports neuronal health by modulating microglial and astrocytic activity [22]. It acts as a histone deacetylase (HDAC) inhibitor, correcting histone acetylation dysregulation in AD, while reducing proinflammatory cytokines and inflammation [23,24]. Lauric acid, on the other hand, enhances mitochondrial function, reduces oxidative damage, and modulates inflammatory pathways [25,26,27]. It has been shown to decrease Aβ deposition and neuroinflammation, further supporting neuronal and glial health. Compared to other natural compounds, these fatty acids uniquely combine epigenetic modulation (via HDAC inhibition) and metabolic support through ketone production, offering a multifaceted approach to AD therapy. Since neurons, microglia, and astrocytes are crucial for maintaining brain function but are disrupted in AD, understanding how butyrate and lauric acid influence these cells presents a promising approach to mitigating neuroinflammation and slowing disease progression. This review explores the mechanisms through which these compounds offer neuroprotection, particularly in modulating neuronal, microglial, and astrocytic activity within the context of AD.

## 2. Alzheimer’s Disease

AD is an irreversible neurodegenerative disorder, accounting for up to 70% of dementia cases in Australia [28]. It affects approximately one in ten Australians aged 65 and three in ten aged 85, although it is not a normal part of aging [29]. AD can occur in two forms: sporadic, the most common type, typically developing after the age of 65, and familial, which is caused by mutations in the APP, PSEN1, and PSEN2 genes [30]. Although rare, familial AD often appears in individuals in their late 40s or early 50s and is clinically indistinguishable from the sporadic form [31]. AD is classified as mild, moderate, or severe based on symptom severity. In the preclinical stage, there are no clinical symptoms [32]. Mild cognitive impairment (MCI), which can progress to AD, involves memory and thinking issues that are beyond what is normal for a person’s age but do not affect independence [33]. In the final stage of AD, memory loss, word-finding issues, and visual/spatial problems impair independence [34].

The accumulation of Aβ peptides and NFTs triggers a cascade of events, including inflammatory responses, oxidative stress, and neuronal damage. Aβ peptides, produced from the abnormal cleavage of amyloid precursor protein (APP), result in impaired clearance of amyloid beta, which leads to excessive Aβ accumulation, resulting in the formation of toxic aggregates and extracellular plaques that drive disease progression [35]. These Aβ aggregates not only impair synaptic function and disrupt neuronal signalling but also serve as potent activators of glial cells, particularly microglia and astrocytes [36]. Tau pathology emerges when hyperphosphorylated tau disrupts normal microtubule function, forming paired helical filaments (PHFs) and NFTs that contribute to neurodegeneration. This process is regulated by an imbalance of kinases and phosphatases, which promotes tau–tau interactions [37]. Hyperphosphorylated tau released into the extracellular space further aggravates pathology by activating glial cells, including microglia and astrocytes, and triggering chronic neuroinflammation [38]. Thus, tau and Aβ pathology not only cause direct damage to neurons but also induce glial cell dysfunction, which amplifies inflammation and accelerates disease progression.

## 3. Role of Microglia in Alzheimer’s Disease

Microglia originate from primitive progenitors in the yolk sac. They migrate into the CNS during early embryogenesis [39]. Microglial migration occurs with yolk sac-derived macrophage progenitor cells entering the CNS via the bloodstream shortly after the embryonic vascular system is established [40]. As highly dynamic cells, microglia perform critical surveillance and homeostatic functions in the CNS, exhibiting phenotypic plasticity that allows for diverse functional responses [41]. They continuously monitor the environment, performing critical surveillance functions and responding to changes in neuronal activity and health. One of their key roles is synaptic pruning, where microglia eliminate weak or redundant synapses during development and in response to activity changes, refining neural circuits for efficient communication between neurons [42].

In their inactive state, microglia detect pathogens and host-derived molecules, such as pathogen-associated molecular patterns (PAMPs) and danger-associated molecular patterns (DAMPs) [43]. This is facilitated by pattern recognition receptors (PRRs) on their surface, making them highly adept at recognizing potential threats [44]. Additionally, microglia facilitate debris clearance by phagocytosing dead neurons and other cellular debris, which is essential for maintaining a healthy extracellular environment and preventing neuroinflammation [45]. Upon activation by invading pathogens, they undergo morphological changes and become immunologically active, participating in phagocytosis, antigen presentation, and communication between innate and adaptive immune systems [46]. The activation of microglia releases inflammatory mediators, including cytokines [47], chemokines, inducible nitric oxide synthase (NOS) [48,49], cyclooxygenase-2 (COX-2) [50,51], and free radicals, such as ROS [52], that disrupt neuronal processes and cellular function. As a result of their activation, microglia also produce a variety of neuroprotective factors that protect neurons from damage, including brain-derived neurotrophic factor (BDNF), glial cell-derived neurotrophic factor (GDNF), and nerve growth factor (NGF) [53]. Microglia exhibit a dual role in inflammation, where these cells can transition between pro-inflammatory (M1) and anti-inflammatory (M2) phenotypes depending on environmental cues, influencing tissue repair and neurodegenerative processes [54].

### 3.1. Microglia Response to Aβ, Tau, Oxidative Stress, and Neuronal Damage

Microglia are known to induce Aβ toxicity. Aβ aggregates trigger the activation of microglia, resulting in the release of NO, ROS, as well as high immunoreactivity towards activation markers (e.g., MHCII and COX2), pro-inflammatory cytokines (e.g., IL-1, MCP-1, MIP-1α, IL-1β, TNFα, and IL-6), and chemokines (e.g., CCR3, CCR5), which may contribute to neuronal death [55]. Key receptors involved in this process include Triggering receptor expressed on myeloid cells 2 (TREM2), which enhances microglial phagocytosis and survival in the presence of Aβ [56]. Aβ-induced activation of microglia leads to the production of ROS through the NADPH oxidase system and reactive nitrogen species (RNS) via the induction of inducible nitric oxide synthase (iNOS) [52]. These toxic factors contribute to amyloidogenesis, impede Aβ clearance, which gradually leads to neurotoxicity. However, there is evidence suggesting that activated microglia can also have neuroprotective effects. For instance, one study demonstrated that overexpression of IL-1β in the APP/PS1 mouse model decreased Aβ plaque load by promoting an increase in activated microglia [57]. Similarly, another study found that IL-6 overexpression in APP transgenic mice reduced Aβ plaque accumulation through the induction of gliosis and microglial activation [58] (Figure 1).

For decades, research has primarily focused on the contribution of Aβ to neuroinflammation, while the link between microglia and tau pathology has received less attention. Upon activation, microglia can facilitate tau spread through extracellular vesicles and synaptic connections, leading to a more widespread pathology. While microglia attempt to clear tau aggregates, their activation can exacerbate tau pathology [59]. A prominent tauopathy mouse model (P301S) showed that immunosuppressants could reduce microglial activation and tau pathology, suggesting that microgliosis may precede tau pathology [60]. However, some studies indicate that tau propagation can trigger microglial activation [61,62]. Recent findings reveal that tau activates the NLRP3 inflammasome, which contributes to inflammatory processes, and that aging leads to microglial dysfunction, increasing their toxicity and altering their response [63]. This aging microglial state may promote Alzheimer’s progression, but eliminating these cells and repopulating with healthier microglia shows promise as a therapeutic strategy, as evidenced by studies demonstrating improved outcomes in both tau and Aβ models. Neuronal death triggers the release of hyperphosphorylated tau aggregates into the extracellular environment, activating microglia and initiating an inflammatory response [4]. In various transgenic models of tauopathy, researchers have documented age-dependent microglial activation and neuroinflammatory changes in CNS structures associated with tau pathology, even without significant neuronal loss [60,64,65,66]. The misfolding of tau, as observed in AD, leads to morphological alterations in microglia, shifting them toward an inflammatory profile characterized by increased expression of pro-inflammatory cytokines such as IL-6 [67]. In response to tau pathology, microglial cells phagocytize aggregated tau and facilitate its transmission to neurons via exosomes. Pharmacological depletion of microglia and inhibition of exosome synthesis have been shown to disrupt tau propagation, highlighting the critical role of microglia in tau dissemination and suggesting their potential as therapeutic targets [68]. Additionally, truncated tau mutations in microglial cultures and mouse models induce the upregulation of pro-inflammatory cytokines through NF-κB and MAPK signalling pathways [69]. Furthermore, pathological tau activates the inflammasome within microglia, a component linked to neuroinflammation and amyloid accumulation [70].

### 3.2. Age-Related Changes in Microglia

Aging is one of the major risk factors for AD. Aging also significantly alters microglial functionality, which impacts their ability to perform these essential tasks effectively. As individuals age, microglia exhibit notable morphological changes, such as reduced branching and smaller cell bodies. These alterations are associated with a decline in key functions, including the clearance of Aβ plaques—a process critical for preventing neurodegenerative diseases [71]. Aged microglia show diminished motility, slower migration, and reduced phagocytic capabilities, particularly in clearing Aβ and other neurotoxic debris [11]. This functional decline is compounded by an increased presence of pro-inflammatory markers and decreased levels of neuroprotective factors, resulting in an overall reduced neuroprotective capacity [72].

Research has documented that aging leads to less evenly distributed microglia in the cortex, with smaller and less symmetrical dendritic arbores. In the retina, aged microglia are smaller, exhibit fewer ramifications, and show slower responses to stimuli, such as ATP [73]. These changes suggest deficits in both initial responses and resolution signalling, contributing to an adverse environment for neuronal health. For instance, aging induces a para-inflammatory state in the retina and increased autofluorescent lipofuscin deposits, which further exacerbate microglial dysfunction [74]. Contrary to earlier assumptions that age-related changes in microglial morphology indicated heightened activation, more recent research suggests these changes reflect dystrophy and senescence rather than activation [75]. Aged microglia often display cytoplasmic inclusions, reduced process complexity, and membrane blebbing, pointing to a diminished capacity to respond to injury rather than an overactive state. This senescence may be driven by factors such as oxidative stress, telomere attrition, and chronic activation [76]. Further studies have revealed that aging impacts microglial signalling pathways, such as the CX3CL1-CX3CR1 axis, crucial for migration, and CD200 levels, which regulate microglial quiescence [77]. The signalling between CX3CL1 and CX3CR1 decreases with age, affecting microglial responses to both basal and inflammatory conditions. Additionally, age-related declines in receptor expression and motility further reflect this senescence [78].

## 4. Role of Astrocytes in Alzheimer’s Disease

Astrocytes are the predominant glial cells in the CNS, essential for diverse functions such as neurovascular coupling and maintaining the integrity of the BBB. They regulate cerebral blood flow, facilitate synaptic formation, and secrete growth factors like GDNF, vascular endothelial growth factor (VEGF), crucial for supporting BBB integrity [79]. In younger individuals, these cells typically have long, slender processes, but as individuals age, these processes become shorter and thicker [80]. Additionally, the density of astrocytes varies by brain region, with increases in density reported in the human cortex and hypothalamus with age [81]. Astrocytes, typically recognized for their supportive functions within the brain, undergo a transformation into reactive states in response to pathological changes, accompanied by significant morphological, molecular, and functional alterations. These reactive astrocytes polarize into distinct phenotypes known as A1 (pro-inflammatory) or A2 (anti-inflammatory) [82].

In pathological states, they may become activated and release pro-inflammatory cytokines (IL-1β, TNF-α, IL-6, and CCL2) that exacerbate neuroinflammation. In contrast, under certain conditions, they also produce anti-inflammatory cytokines like IL-10 and TGF-β, promoting tissue repair and mitigating damage [83]. Moreover, reactive astrocytes secrete CCL2, a chemokine that recruits peripheral immune cells, such as monocytes and T-cells, to sites of neuroinflammation, further amplifying the inflammatory environment (Palmer & Ousman, 2018) [84]. The secretory profile of aged astrocytes is also altered, with increased production of cytokines like CXCL10 and CXCL5. These cytokines are involved in immune cell recruitment and inflammatory responses [84,85]. Astrocytes are a key source of ROS, particularly under stress, injury, or inflammatory conditions [86]. Astrocytes play a key role in regulating synaptic function by taking up and recycling neurotransmitters like glutamate and releasing gliotransmitters that modulate synaptic activity [87].

### 4.1. Astrocytes’ Response to Aβ, Tau, and Neuronal Damage

Astrocytes respond to neuroinflammation by tightening junctions between endothelial cells, limiting immune cell entry into the brain. However, chronic activation can lead to tissue damage [10]. In AD, astrocytes exhibit a dual role where they contribute to the clearance of Aβ plaques, but their inflammatory activation can paradoxically increase Aβ production [88]. Microglia activate astrocytes through signaling molecules like IL-1α, C1q, and TNF-α, which stimulate enzymes involved in Aβ production, thereby potentially worsening neuronal Aβ accumulation. This inflammatory activation of astrocytes suggests a feedback loop in Aβ production, highlighting their complex involvement in the pathogenesis of AD [89]. Furthermore, astrocytes influence tau protein metabolism, potentially contributing to tau pathology through processes such as internalization and degradation, which can be disrupted in a state of inflammation [90].

In AD, neuroplasticity is impaired, disrupting neuronal signalling and leading to cognitive and motor deficits [91]. This is primarily due to the oxidative environment created when activated astrocytes release NO and ROS, which together form peroxynitrite and other potent neurotoxins [90]. In AD, reactive astrocytes show impaired neurotransmitter handling, which contributes to glutamate excitotoxicity. The excessive build-up of glutamate at synapses causes an abnormal influx of calcium, leading to mitochondrial dysfunction and ultimately neuronal death [92]. Astrocytes interact with microglia to regulate synapse elimination can become pathological, resulting in excessive synaptic pruning and contributing to cognitive decline [93].

The induction of A1 astrocytes, triggered by microglial cytokines, correlates with neurotoxic properties observed in AD. Reactive astrocytes in AD also show changes in calcium signaling and gene expression, including reduced levels of IP3 receptor type 2, which are associated with early AD pathogenesis and Aβ accumulation [94]. Astrocytes additionally influence AD-related protein metabolism by facilitating Aβ plaque degradation and internalizing Aβ1–42 peptides, although under inflammatory conditions, they can increase Aβ production. The presence of the APOE-ε4 allele, a significant risk factor for AD, alters astrocytic function, promoting inflammation and synaptic loss [95,96]. Moreover, astrocytes are involved in tau protein metabolism, potentially contributing to tau pathology through processes such as internalization, degradation, and potential propagation. In states of neuroinflammation and oxidative stress, astrocytes release proinflammatory cytokines and ROS, exacerbating neuronal damage. Conversely, they produce antioxidants like glutathione (GSH) to counteract oxidative stress, although this protective function can be compromised with prolonged exposure to amyloid [97,98] (Figure 1).

### 4.2. Age Related Changes in Astrocytes

Although aged astrocytes enhance their expression of antioxidant defenses like peroxisome proliferator-activated receptor-gamma coactivator 1-alpha (PGC-1α) and nuclear factor erythroid 2-related factor 2 (Nrf2), these mechanisms are often insufficient to counteract the extensive oxidative damage associated with aging [86]. In response to the effects of aging-associated proteases, aged astrocytes upregulate Serpina3n, a serine protease inhibitor [99]. Additionally, there is a dysregulation in cholesterol synthesis, with reduced expression of HMG-CoA reductase and increased expression of cholesterol transport receptors in aged astrocytes. This disruption in cholesterol homeostasis may impact neuronal function. Epigenetic changes also occur with aging, including both hyper- and hypo-methylation of DNA and alterations in histone modifications [100]. In neurological disorders characterized by BBB disruptions, astrocytes respond by tightening junctions between endothelial cells, thereby limiting the entry of immune cells into the brain. Depending on the stage of disease progression, astrocytes can either exacerbate inflammation and tissue damage or promote tissue repair [101].

At the molecular level, aging astrocytes show increased expression of glial fibrillary acidic protein (GFAP), indicative of reactivity [102]. Moreover, aging astrocytes exhibit elevated levels of complement system components such as C3 and C4B. This suggests an increased involvement of astrocytes in inflammatory responses and potential contributions to cognitive decline through the modulation of synaptic connections [84]. Similarly, there is an upregulation of Major Histocompatibility Complex I (MHC I) in aged astrocytes, which could enhance antigen presentation and contribute to chronic low-level inflammation and cognitive impairment [103].

Gut microbiotas communicate bidirectionally with astrocytes, influencing their activity. Dysbiosis can modulate astrocytic responses, exacerbating neuroinflammation [104]. SCFAs can enhance astrocyte maturation and function, promoting anti-inflammatory responses. Additionally, systemic inflammation from the gut may stimulate the release of astrocytes, worsening CNS inflammation [105].

## 5. Neuronal Activity in Alzheimer’s Disease

Neurons are the essential components of the nervous system, developing from neural stem cells during embryonic stages. They differentiate into various types, including sensory, motor, and interneurons, each with specialized functions [106]. Neurons communicate via electrical impulses and neurotransmitter release, facilitating reflexes, sensory perception, and complex cognitive processes [107].

### Neurons’ Response to Aβ, Tau, and Oxidative Stress

In response to Aβ accumulation, neurons activate microglia and astrocytes to help clear Aβ deposits. However, excessive Aβ can impair synaptic function and trigger neuroinflammation, contributing to cognitive decline in AD [108]. As discussed under AD pathology, hyperphosphorylated tau protein forms NFTs, leading to neuronal dysfunction and death. Although neurons attempt to stabilize microtubules in response to tau pathology, they can ultimately surrender to stress, furthering neurodegenerative processes.

In AD, neuronal dysfunction and degeneration play pivotal roles in the progression of cognitive decline and memory loss. NFTs, intracellular aggregates of hyperphosphorylated tau protein, disrupt neuronal structure and function. Concurrently, extracellular deposition of Aβ protein forms amyloid plaques, further impairing synaptic transmission and neuronal viability [98]. Synaptic dysfunction, exacerbated by tau-mediated microtubule destabilization and beta-amyloid-induced oxidative stress, underlies early cognitive deficits. Neuronal loss, particularly in the hippocampus and neocortex, correlates with disease severity and manifests as widespread neuroinflammation [109,110]. Mitochondrial dysfunction and dysregulated calcium homeostasis contribute to neuronal vulnerability and apoptotic pathways. Elucidating these molecular and cellular mechanisms is critical for developing targeted therapies to mitigate neurodegeneration and improve cognitive outcomes in AD patients [111,112].

The neurovascular unit (NVU), comprising neurons, astrocytes, microglia, and endothelial cells, maintains brain homeostasis by regulating blood flow, nutrient delivery, and waste clearance. In AD, breakdown of the NVU and compromised BBB integrity led to impaired cerebral blood flow and neuroinflammation, further compromising neuronal health [113,114]. Glial cells, including astrocytes and microglia, become reactive in response to AD pathology, releasing pro-inflammatory cytokines and exacerbating neuroinflammation [115]. This inflammatory environment contributes to synaptic dysfunction and neuronal loss, perpetuating the cycle of neurodegeneration seen in AD [116].

## 6. Microbiota–Gut–Brain Axis Connection

Research indicates that gut microbiota play an important role in bidirectional communication between the gut and nervous system [117]. In addition to regulating brain chemistry, there is growing evidence that the intestinal microbiota influences brain-gut interactions at different points in time and in different systems [118]. One of the affected systems is the neuro-endocrine system, which is involved in stress response, anxiety, and memory [119].

The mechanisms underlying gut–brain axis (GBA) communications involve neuro-immuno-endocrine mediators. Several systems are involved in this axis, including the autonomic (ANS) and enteric nervous systems (ENS), the endocrine system, the hypothalamus–pituitary–adrenal axis (HPA), the immune system, as well as the microbiota and their metabolites. The ANS, with its sympathetic and parasympathetic limbs, transmits afferent signals from the intestinal lumen to the CNS, and efferent signals vice versa. As the organism’s core efferent axis to stress, the HPA axis coordinates its adaptive response to all types of stress. As part of the limbic system, it is primarily responsible for memory and emotional responses [120]. Stress and elevated systemic pro-inflammatory cytokines activate this system, which stimulates the pituitary gland to secrete adrenocorticotropic hormone (ACTH), which, in turn, releases cortisol from the adrenal glands by secreting corticotropin-releasing factor (CRF) from the hypothalamus [121] (Figure 2).

Despite individual differences in microbiota profiles, there is a common distribution and abundance of bacterial phylotypes along the human gastrointestinal tract [122]. The two most common phyla accounting for the microbiome are *Firmicutes* and *Bacteroides* [123]. Various clinical and experimental studies suggest that enteric microbiota play an important role in GBA, interacting not only with intestinal cells and ENS but also directly with the CNS [124].

Researchers have discovered significant roles of the gut microbiome in many aspects of brain development, including myelination, neurogenesis, and microglia maturation [125]. In addition to contributing to neuroinflammatory and psychiatric disorders [126,127,128], gut bacteria also contribute to brain development [129]. In addition to synthesizing neurotransmitters such as GABA, noradrenaline, and dopamine, the gut microbiota modulates the immune system, produces metabolites such as SCFAs, metabolizes essential amino acids such as tryptophan, and activates NGF, GDNF, and BDNF, which have implications for neurological disorders.

Growing evidence shows that dysbiosis of the gut microbiota leads to increased gut permeability and immune activation, resulting in systemic inflammation and neuroinflammation, brain injury, and ultimately neurodegeneration.

## 7. Role of Gut Microbiota in Alzheimer’s Disease Pathology

The gut microbiota, a complex community of microorganisms in the gastrointestinal tract, plays a pivotal role in AD pathology through its interactions with the gut–brain axis [130,131]. Dysbiosis, or microbial imbalance, disrupts this bidirectional communication network, contributing to neuroinflammation, Aβ pathology, and cognitive decline [132]. Microbial metabolites like SCFA, essential for brain health, are often important for gut health/dysbiosis, brain homeostasis, and reducing Aβ deposition [133]. Additionally, pro-inflammatory molecules such as LPS from gram-negative bacteria can cross the BBB, triggering neuroinflammatory responses and activating microglial cells, which exacerbate neuronal damage. Age-related reductions in microbial diversity and sex-based differences in microbiota composition further influence susceptibility to AD [134]. There are significant differences between the gut microbiomes of MCI patients and cognitively normal subjects, highlighting specific gut microbiome signatures associated with MCI as well as correlations with cerebrospinal fluid tau levels, total tau levels, and phosphorylated tau levels [135]. Gut flora activation of NLRP3, a protein that plays a key role in the body’s immune response and inflammatory process, has also been shown to play a role in AD pathogenesis. NLRP3 inflammasome activation after faecal microbiota transplantation (FMT) from AD patients causes systemic inflammation and neuroinflammation in the hippocampus in animal models [136]. Gut microbiota impact on AD risk has been discussed primarily through active metabolites and signalling molecules, including trimethylamine N-oxide (TMAO), bile acids, dysregulated P-glycoproteins, and microRNA-146a, which is sensitive to nuclear factor-B [137].

## 8. Short Chain Fatty Acids

Dietary fibers are resistant starch that cannot be hydrolyzed by endogenous enzymes, but microbiota in the colon convert dietary fiber into SCFAs [138]. SCFAs are five carbons or less, including valeric acid and isovaleric acid (C5), isobutyric acid and butyric acid (C4), propionic acid (C3), acetic acid (C2), and formic acid (C1). SCFAs help overcome brain hypometabolism, which contributes to neuronal dysfunction in AD and other neurodegenerative conditions. SCFAs may also modulate microglial maturation and reduce neuroinflammation, both of which are important across a variety of neurodegenerative disorders, including AD [139]. Comparing healthy subjects to AD patients, a microbiome analysis revealed a lower prevalence of bacteria synthesizing butyrate and higher levels of proinflammatory bacteria [126]. Butyrate crosses the blood–brain barrier (BBB) more effectively than other short-chain fatty acids (SCFAs).

### 8.1. Butyrate

#### 8.1.1. Origin, Structure, and Mechanism of Action

Butyrate is an SCFA with the chemical formula C_4_H_8_O_2_, consisting of a four-carbon chain with a carboxylic acid functional group. It is primarily produced through anaerobic fermentative processes, like other SCFAs, and naturally present in high concentration in the lumen of the large intestine [140]. Key bacterial species, such as *Faecalibacterium prausnitzii*, *Roseburia*, and *Eubacterium rectale*, are instrumental in converting dietary fibers into butyrate [141]. Two acetyl CoA molecules are shortened into acetoacetyl CoA, which is converted through L (+)-beta-hydroxybutyryl CoA and crotonyl CoA intermediates to butyryl CoA. Butyryl CoA is then converted to butyrate either by butyrate kinase or butyryl CoA and acetate CoA transferase [142]. Food sources, such as milk, legumes (beans, peas, and soybeans), fruits, nuts, cereals, and whole grains, also supply butyrate [143]. Research indicates that butyrate can cross the BBB, although the exact transporter involved remains unclear [144]. In vitro studies have reported that butyrate inhibits the production of pro-inflammatory cytokines, NF-κB activation, and nitric oxide, but the exact effects seem to vary depending on the type of cell, stimulus, and concentration of butyrate [145,146,147,148,149].

Butyrate is a histone deacetylase (HDAC) inhibitor (Table 1). It is mediated through inhibition of a broad histone deacetylase (HDAC) inhibitor, inhibiting histone and non-histone isoform classes I and II. Ensuing hyperacetylation of transcription factors alters gene expression patterns by altering both histone and non-histone proteins [150]. As in AD, acetylation homeostasis is uncontrolled, butyrate is considered a potential therapeutic in AD as it acts as an HDAC inhibitor [23]. Butyrate modulates neuroinflammation and promotes brain health by binding to FFAR2 and FFAR3 receptors to regulate the immune systems. The effect on cytokine pathways is unknown [151]. In CNS diseases, butyrate exerts neuroprotective effects such as anti-inflammatory [152], antioxidant, anti-apoptotic [153], and cognitive improvements [154].

#### 8.1.2. Effects on Microglia, Astrocytes, and Neurons

Research has shown that butyrate may shift microglia from a pro-inflammatory (M1) phenotype to an anti-inflammatory (M2) phenotype, promoting neuroprotection and reducing neuroinflammation. By inhibiting the overactive immune response of microglia, butyrate could mitigate the neuroinflammatory component of AD [155]. Furthermore, administration of butyric acid-producing *Clostridium butyricum* to APP/PS1 mice prevented microglia activation, cognitive impairments, Aβ deposition, and transient dysbiosis of the intestinal flora. Furthermore, butyrate treatment also decreased the expression of inflammatory cytokines, such as IL-1β, IL-6, and TNF-α. Additionally, butyrate treatment also decreased the formation of Aβ plaques. Further research has revealed that butyrate treatment inhibits microglial activation by reducing NF-κB p65 phosphorylation in Aβ-induced BV2 microglia, and it lowers the expression of CD11b and COX-2 in these cells [156]. In a study on mice with chronic alcohol-induced impairment, sodium butyrate administration helped correct microglial polarity imbalances (M1/M2) through interaction with the G protein-coupled receptor (GPCR)109A receptor. This interaction led to increased PPAR-γ expression and reduced activation of the TLR4/NF-κB pathway. Additionally, sequencing of mouse gut microbiota revealed that sodium butyrate treatment decreased the abundance of genera such as *Bifidobacterium*, *Bacteroides*, and *Lactobacillus*, while increasing *Parvibacter*, *Faecalibaculum*, and *Alloprevotella* [157]. In another study, the bacterium *Agathobaculum butyriciproducens* (SR79), a strict anaerobe that produces butyric acid, was shown to improve cognitive function and reduce AD pathology in animal models by modulating neuroinflammation and IGF-1 signalling [158]. In microglia, sodium butyrate can upregulate the PI3K/AKT/CREB/BDNF signaling pathway, which contributes to long-term potentiation and synaptic plasticity [159].

Studies have shown that sodium butyrate can enhance the mitochondrial function of astrocytes [160,161]. MCT1, a monocarboxylic acid transporter, is expressed on astrocytes, which provides a mechanism for sodium butyrate’s effects on these cells. Notably, studies have highlighted that CPT1a, a key enzyme involved in fatty acid utilization, can be activated by sodium butyrate, thereby promoting fatty acid oxidation. Furthermore, the expression of various CPT1A subtypes and the presence of neurons in astrocytes suggest that these cells have significant potential for fatty acid oxidation in the brain [144]. The APOE genotype also influences astrocyte lipid utilization. Research has found that APOE4 astrocytes exhibit distinct metabolic patterns compared to other APOE variants [162]. Given that APOE4 is a known risk factor for AD, studying the effects of sodium butyrate across different APOE phenotypes is crucial. Understanding how butyrate influences astrocyte metabolism could provide valuable insights into its potential therapeutic applications.

In an in vitro study using Aβ-induced N2a cells, it was shown that sodium butyrate promotes mitochondrial function and cell proliferation [163]. Researchers found that sodium butyrate protected rat brains against amphetamine-induced oxidative stress [164]. Aβ-mediated oxidative stress is a major factor in the pathology of AD sodium butyrate is considered to provide neuroprotection [165]. A mouse model of AD shows that sodium butyrate ameliorates cognitive impairment during the early and advanced stages of the disease [23]. In an APPPS1-21 mice amelioration of cognitive impairment was associated with an increase in the expression of memory-consolidation genes, such as MYST4, Marcksl1, GluR1, SNAP25, and SHANK3, along with elevated hippocampal acetylation [166]. As sodium butyrate reversed radiation-induced decreases in (BDNF)/phosphorylated cyclic adenosine monophosphate (cAMP) response element binding protein (pCREB) expression, it was identified as a potential mechanism by which sodium butyrate enhanced cognition [167]. Sodium butyrate promotes long-term potentiation and synaptic plasticity in microglia by upregulating the Phosphatidylinositol 3-kinase (PI3K)/protein kinase B (AKT)/cAMP response element-binding protein (CREB)/BDNF pathway [159]. Preclinical studies have demonstrated that 40 mM sodium butyrate supplementation increases Aβ plaque deposition or the aggregation kinetics of Aβ in germ-free APP/PS1 mice [168].

Acetylation homeostasis is compromised in AD, leading to the dysregulation of epigenetic mechanisms in the cells [23]. Dysregulation of histone acetylation is linked to the severe amyloid pathology [169]. A 12-week feeding schedule incorporating sodium butyrate improved cognitive function, associative memory, and Aβ42 deposition in male transgenic mice expressing 5 familial AD 126 mutations (5xFAD) in mice [23]. Furthermore, butyrate lowered the secretion of proinflammatory cytokines by microglia and improved the associated neuroinflammation [168]. Butyrate exhibits anti-inflammatory properties by inhibiting the expression of cyclooxygenase-2 (COX-2) in BV2 microglial cells exposed to Aβ) This inhibition is accompanied by a decrease in the phosphorylation of NF-κB-p65, a critical step in the activation of the NF-κB signalling pathway, which regulates the production of inflammatory mediators. By targeting these mechanisms, butyrate helps to suppress inflammation [156].

There is evidence that butyrate maintains endothelial integrity in mice and can increase the expression of tight junction proteins in their hippocampus and frontal cortex [170]. By connecting transmembrane tight junction proteins, endothelial cells form the BBB. Research suggests that BBB dysfunction contributes to the deposition of Aβ by activating BACE1 and γ-secretase, thus increasing the conversion of APP into Aβ and by preventing normal transport through the BBB [171]. The presence of SCFAs is essential for the establishment of a normal BBB and during disease progression for the protection and repair of the BBB. Treatment with sodium butyrate can reduce BBB permeability in germ-free mice [172]. However, further research on butyrate’s effect on the BBB in AD is warranted. In addition, in vito studies have exhibited the capacity of butyrate to inhibit the self-assembly of Aβ40 and Aβ42 monomers into Aβ fibrils [173]. In a study of older Italians with cognitive impairment, SCFA butyrate was associated with decreased brain amyloid deposition based on florbetapir amyloid PET and blood samples [174]. 

Delivering butyrate to the CNS presents several challenges. One of the primary obstacles is the BBB, a selective structure that tightly regulates what enters the brain from the bloodstream. Although butyrate can cross the BBB through transporters like MCTs or via passive diffusion, the efficiency of this process is not well established, and the concentrations that reach the brain are often suboptimal [144]. Moreover, butyrate is volatile in the gut and bloodstream, where it is rapidly metabolised or degraded, further limiting its potential therapeutic effect. Oral administration of butyrate is especially problematic, as it has potentially low bioavailability [175]. To address these challenges, researchers are exploring novel strategies, including encapsulation techniques or combining butyrate with other molecules to enhance its stability and bioavailability. Such formulations may improve their delivery to the CNS, but more research is needed to optimize these approaches [176].

#### 8.1.3. Therapeutic Potential, Challenges, and Limitations of Butyrate Delivery to the CNS

The therapeutic potential of butyrate can be significantly enhanced when combined with other agents, making it a promising candidate for adjunctive therapies in AD. Combining these compounds with standard AD therapeutics or anti-inflammatory agents may provide synergistic effects. However, interactions with amyloid-targeting drugs or anti-diabetic medications must be studied, as unintended metabolic or immune responses could occur. Butyrate, with its anti-inflammatory and neuroprotective properties, may have synergistic effects when used alongside other therapeutic agents. For instance, anti-inflammatory drugs such as nonsteroidal anti-inflammatory drugs (NSAIDs) or corticosteroids, which are used to reduce neuroinflammation, may see enhanced efficacy when combined with butyrate [177]. As mentioned before, butyrate itself can modulate the immune response, further supporting its anti-inflammatory action in the brain. Similarly, antioxidants like resveratrol or curcumin, which are known for their neuroprotective effects, may work synergistically with butyrate to combat oxidative stress, a hallmark of many neurodegenerative diseases [178]. Combining butyrate with probiotics or prebiotics—which modulate the gut microbiota—can also enhance butyrate production in the gut, potentially improving both gut and brain health [179].

Butyrate can influence host metabolism indirectly by interacting with the gut–brain axis. For example, butyrate may enhance the proportion of cholinergic enteric neurons through epigenetic mechanisms [180,181]. Additionally, butyrate can cross the blood–brain barrier, activating the vagus nerve and hypothalamus, which in turn affects appetite and eating behaviour [182,183]. Some of butyrate’s metabolic benefits are thought to be mediated through gluconeogenesis in the gut epithelium and through gut–brain neural circuits that improve insulin sensitivity and glucose tolerance. For instance, butyrate binds to receptors on intestinal cells, signalling the brain through the cAMP pathway [184].

Dietary fiber intake is a major determinant of butyrate levels, as fiber serves as a substrate for microbiota fermentation. A diet rich in fibers like resistant starch and prebiotics supports the growth of butyrate-producing bacteria, leading to higher levels of this SCFA in the gut [185]. However, gut dysbiosis, an imbalance in microbiota composition, can result in decreased butyrate production, which may contribute to systemic inflammation and neurodegeneration [186]. AD, dysbiosis, or an imbalance in the gut microbiota, has been associated with a reduction in butyrate production, which in turn contributes to disease progression. For example, in AD, the abundance of butyrate-producing bacteria is often reduced, further exacerbating disease progression. Conversely, interventions aimed at restoring a healthy gut microbiota, such as probiotics, prebiotics, or dietary changes, can increase butyrate production and improve both gut and brain health [187,188]. Thus, the gut microbiota’s composition plays a central role in determining butyrate levels and, by extension, its therapeutic potential.

Modulating the gut microbiota is a promising approach to enhance butyrate production and improve therapeutic outcomes in diseases like AD [189]. Other strategies, such as faecal microbiota transplantation (FMT), are also being investigated to directly alter the gut microbiota and boost butyrate production [190]. As research in this area continues to evolve, these microbiome-based therapies hold promise as adjuncts to conventional treatments for neurological disorders like AD.

**Table 1 nutrients-17-02286-t001:** Summary of effects of butyrate in Amyloid Aggregation, Inflammation, Oxidative Stress, and Cellular Integrity.

	Treatment	Model	Main Finding	Author
Aggregation of AB	Levels of butyrate amongst 89 older people with cognitive performance from normal to impaired	Human	Butyrate was associated with decreased brain amyloid deposition.	[174]
	Addition of NaB (2 mM) to cell culture.	Mouse neuroblastoma cells (N2a)	Supressed expression of APP and promoted effect of Neprilysin (NEP)	[163]
	Addition of butyrate to Aβ40 and Aβ42 monomers at 0:1, 1:1, and 4:1 SCFA: Aβ molar ratio.	In vitro	Butyrate inhibited the self-assembly of Aβ40 and Aβ42 monomers into Aβ fibrils.	[173]
Pro-inflammatory mediators	Butyrate was added to cell cultures at 0.2, 2, and 20 mmol/L	Human monocytes	Inhibited IL-10 production in LPS stimulated human monocytes and MCP1 in both LPS and non-LPS stimulated human monocytes.	[191]
	Butyrate was added to cell cultures at 0.2, 2, and 20 mmol/L	Human peripheral blood mononuclear cells (PBMC)	inhibited TNF-α and IFN-y secretion in human PBMC	[191]
	Oral treatment of 100 mM of sodium butyrate for 3 weeks.	Specific pathogen free (SPF) C57BL/6 mice	NaB promotes the expansion of Foxp3+ regulatory T cells	[192]
	Cell lines treated with Butyrate on its own and in combination with other SCFAs	Human THP-1 monocytic cell line	Reduced secretion of MCP-1, compared to cells treated with control and significantly reduced the secretion of IL-1β in human THP-1 monocytic cell line	[193]
	Mice were administered streptomycin (5 g/L) containing water and inoculated with 1 × 10^9^ CFUs of Enterotoxigenic *Escherichia coli* (ETEC). Mice received sodium Butyrate (5 g/L) via water one day before streptomycin and throughout experiment.	Male GPCR109A+/+ and GPCR109A−/− mice	sodium butyrate reversed the increased expression of proinflammatory cytokines IL-1β, IL-6, and TNF-α in GPCR109A+/+ mice but showed no reversal effect in GPCR109A−/− mice	[194]
Reactive Oxygen Species	Addition of NaB (2 mM) to cell culture.	Mouse neuroblastoma cells (N2a)	NaB Inhibited the production of Aβ induced ROS	[163]
Histone acetylation	Cell lines were incubated in presence of 10 mmol/L sodium butyrate	Human breast cancer MCF—7 cell line	Sodium butyrate inhibits histone deacetylase	[195]
	Nab was dissolved in 0.01 M phosphate buffer saline (PSB) and administered daily at a final concentration of 1.2 g/kg of body weight.	APPPS1—21 double transgenic mice that co-express KM670/671NL mutated amyloid protein precursor (ABPP) and L166P mutated presenilin 1 (PS1)	Enhanced associative memory and elevated hippocampal acetylation at H3K14, H4K5, and H4K12 sites.	[166]
	Nab was added to chow pellets and was administered at either 5 mh/kg/day, or 15 mg/kg/day for 12 weeks.	Male transgenic mice expressing 5 familial AD mutations [APP: K67ON/M671 L (Swedish) + 1716 V (Florida) + V7171 (London) and PS1: M146 L + L286V], crossed with APOE ε3	Improved associative memory and cognitive functioning. A 40% decrease in brain Aβ levels.	[23]
Endothelial and colonic epithelial integrity	Nab was administered intraperitoneally at 200 mg/kg body weight.	Male C57BL/6 mice	Increased expression of tight junction (TJ) proteins occluding and ZO-1.	[23]
	Mice were administered streptomycin (5 g/L) containing water and inoculated with 1 × 10^9^ CFUs of Enterotoxigenic *Escherichia coli* (ETEC). Mice received sodium Butyrate (5 g/L) via water one day before streptomycin and throughout experiment.	Male GPCR109A+/+ and GPCR109A−/− mice	Sodium butyrate reversed the decreased expression of TJ proteins Cldn1, Cldn2, Cldn3, Ocln, and Zo-1 caused by ETEC in GPCR109A+/+ mice, but not in GPCR109A−/− mice.	[194]

## 9. Medium Chain Fatty Acids

Medium-chain triglycerides, found in coconut and palm kernel oils, are esters of glycerol and MCFAs. As a result, these oils offer a straightforward, cost-effective, and widely accepted method for consuming MCFAs [196]. Existing literature shows fewer data on MCFAs and their role in human health and disease. MCFAs are known to bind and activate GPCR84, resulting in Ca^2+^ mobilization and inhibition of cAMP [197]. In the presence of lipopolysaccharides, MCFAs specifically stimulate the secretion of proinflammatory cytokines such as IL-12 p40 (IL-12B) subunit. Which induces and maintains Th1-cell responses and inhibits Th2-cell responses, promoting cell-mediated immunity [198]. MCFAs reduce fat deposition in both animals and humans by increasing thermogenesis and fat oxidation [199]. MCFAs decrease oxidative stress markers malondialdehyde (MDA) and inflammation in human liver cells [200].

MCFAs influence the progression and development of AD by affecting modifiable risk factors of AD. In animal models of insulin resistance, MCFAs reverse fatty liver and improve cardiac function [201], while in humans, MCFAs improve glucose disposal [202]. By activating GPCR40, MCFA increases insulin release from pancreatic β-cells in response to glucose. GPCR40, being abundantly expressed in the pancreas and brain, facilitates the reduction of the progression of AD and diabetes. The activation of GRP40 in pancreatic β-cells results in increased cytosolic Ca^+^, which depolarizes the β-cell, leading to increased secretion of insulin. Based on the metabolic effects of ketones on the human brain, it has been suggested that the ketogenic diet may be able to treat metabolic changes underlying AD. In a study, Aβ-induced toxicity in cortical neurons was prevented by MCFAs, and oxidative stress markers were diminished in these cells by coconut oil [203]. Among the MCFAs, lauric acid is gaining interest due to its immune function, anti-inflammatory, improving insulin resistance and reversing steatosis and cardiac dysfunction [202].

### 9.1. Lauric Acid

#### 9.1.1. Origin, Structure, and Mechanism of Action

Lauric acid, a 12-carbon MCFA predominantly found in coconut oil, has emerged as a promising compound for targeting key mechanisms involved in AD. Lauric acid, or dodecanoic acid, has the chemical formula C_12_H_24_O_2_ and is a saturated fatty acid with a 12-carbon chain [202]. Known for its anti-inflammatory [204], antimicrobial [205], and metabolic benefits [206], lauric acid has demonstrated potential in improving brain energy metabolism and modulating the activity of brain-resident immune cells, such as microglial cells and astrocytes [207,208]. Its amphipathic nature allows for mitochondrial membrane penetration and supports ketone body production through β-oxidation in astrocytes [46]. These cells are critical for maintaining brain homeostasis but become dysregulated in AD, contributing to neuroinflammation and neuronal damage (Table 2). By influencing the activity of these brain cells and providing an alternative energy source to the glucose-deprived brain, lauric acid offers a unique, multifaceted approach to addressing the complex pathology of AD [27,209].

Upon ingestion through dietary sources such as coconut oil and dairy products, lauric acid is emulsified in the stomach and small intestine with the help of bile salts and pancreatic enzymes [210]. High doses of lauric acid can influence lipid metabolism and potentially exacerbate lipid disorders [211]. Lauric acid occupies the sn-1 and sn-3 positions of the triglycerides, which makes lipase hydrolysis more effective. The hydrolysis process involves multiple steps. The first two steps yield 1,2-diglycerides and free fatty acids, and then monoglycerides and additional fatty acids. The final hydrolysis step results in the production of monoglycerides, such as monolaurin, which can be absorbed by intestinal cells [212]. Lauric acid is directly absorbed into intestinal cells and transported via the portal vein to the liver. In the liver, lauric acid undergoes rapid metabolism, primarily through β-oxidation, which produces acetyl-CoA for the citric acid cycle. It is also metabolized into ketone bodies, which are transported to other tissues like the brain and muscles for energy [213]. In the bloodstream, lauric acid is rapidly oxidized and metabolized. The likelihood of fat accumulation and the production of ectopic fat metabolites is lower for lauric acid compared to other saturated fatty acids. As a result, LA does not contribute to insulin resistance or inflammation [214]. Lauric acid can diffuse freely across the mitochondrial membrane which require carnitine for transport [215].

#### 9.1.2. Effects on Microglia, Astrocytes, and Neurons

One of the key effects of lauric acid is the modulation of microglial function. Lauric acid has been shown to modulate microglial activation, promoting a shift from a pro-inflammatory to an anti-inflammatory phenotype [208]. These steps involve the activation of pattern recognition receptors, such as toll-like receptors, which trigger the production of inflammatory mediators like free radicals, NO, and proinflammatory cytokines, including TNFα and IL-1β. LPS binds to TLR4, leading to the activation of microglia both in vitro [216] and in vivo [217]. This change may help reduce the neuroinflammation typically observed in AD. Furthermore, lauric acid has been shown to enhance microglial phagocytosis, clearing Aβ plaques [218]. This property indicates that lauric acid might help reduce one of the main pathological features of the disease.

Neuropathology of AD is caused by hyperactive microglia that release NO and proinflammatory cytokines. Research has been conducted on the effects of lauric acid on microglia activated by lipopolysaccharide (LPS) [208]. LPS directly binds toll-like receptors and initiates a signalling cascade that produces proinflammatory cytokines, ROS, and NO [219]. During brain inflammation, TLR4 is upregulated because microglia express TLR1–9 members along with other TLR family members. Lauric acid suppresses LPS-stimulated production of NO, ROS, and proinflammatory cytokines and phagocytosis by microglia. It also mitigates Aβ-induced phagocytosis. In addition to suppressing reactive oxygen species and proinflammatory cytokines produced by LPS, lauric acid also decreases p38-mitogen activated protein kinase phosphorylation and c-Jun *N*-terminal kinase activity [208]. Lauric acid also reduces ERK1/2 phosphorylation and may indirectly inhibit NF-κB translocation via ketone-body-mediated signaling, contributing to its neuroprotective profile [220].

In astrocytes, lauric acid typically increases the expression of GDND, IL-6, and CCL-2. These changes were attributed to lauric acid-induced phosphorylation of extracellular signal-regulated kinase. Lauric acid also enhances the presynaptic protein levels [207]. The ability of lauric acid to convert into ketone bodies and serve as an alternative fuel for the brain assists with bypassing the glucose metabolism dysfunction observed in AD.

#### 9.1.3. Therapeutic Potential, Challenges, and Limitations of Lauric Acid Delivery to the CNS

Preclinical studies have provided strong evidence supporting the efficacy of lauric acid in AD models [26]. Preclinical studies indicate that lauric acid could help address multiple aspects of AD pathology, including amyloid plaque formation and tau pathology, offering a potential therapeutic strategy. Despite its potential, there are several challenges in utilizing lauric acid as a therapeutic agent for AD, particularly in terms of its delivery to the CNS. One of the main obstacles is ensuring efficient penetration of the BBB. While lauric acid is known to cross the BBB more easily than long-chain fatty acids, the concentrations needed for therapeutic efficacy may still be difficult to achieve [221]. Additionally, the bioavailability of lauric acid is influenced by factors such as solubility and absorption in the gastrointestinal tract. Effective delivery strategies, such as nanoemulsion or encapsulation techniques, may be required to enhance the bioavailability and brain uptake of lauric acid [222]. Furthermore, while lauric acid is generally regarded as safe in moderate amounts, its long-term safety at higher doses has not been thoroughly evaluated, and potential side effects, particularly in individuals with neurodegenerative diseases, need further investigation [223]. Long-term studies are essential to determine the safety and tolerability of lauric acid as a therapeutic agent in AD.

Lauric acid’s potential efficacy in AD may be enhanced when used in combination with other therapeutic agents. A potential synergy lies in the use of lauric acid alongside dietary interventions, such as ketogenic diets, which aim to shift brain metabolism from glucose to ketones. Since lauric acid is metabolized into ketone bodies, it could complement ketogenic diets, enhancing the therapeutic effects on brain energy metabolism and cognitive function [209].

**Table 2 nutrients-17-02286-t002:** Summary of effects of lauric acid in Alzheimer’s disease.

Treatment	Model	Main Findings	Authors
Medium chain triglycerides (including Lauric acid)	Healthy adults	MCTs, including Lauric acid, improve cognitive function in healthy adults, suggesting benefits for brain health and potential support against cognitive decline.	[224]
Ketone bodies (including those from Lauric acid)	Transgenic Mouse	Ketone bodies from Lauric acid may contribute to neuroprotection and cognitive enhancement, influenced by dietary fats.	[225]
Ketogenic diet (including Lauric acid)	Mild cognitive impairment	Ketogenic diets, enriched with Lauric acid, improve memory in mild cognitive impairment, potentially impacting Alzheimer’s disease progression.	[226]
Ketogenic diet (including medium-chain triglycerides like Lauric acid)	Mild to moderate Alzheimer’s disease	Ketogenic diets, including Lauric acid, modestly improve cognitive function in Alzheimer’s patients, suggesting therapeutic potential.	[227]

While the neuroprotective potential of butyrate and lauric acid has been well demonstrated in cellular and animal models, their effectiveness in humans remains to be validated. Future research should focus on translational studies and early-phase clinical trials to assess safety, tolerability, and pharmacodynamics in AD patients. These trials should use advanced neuroimaging techniques, cognitive tests, and blood-based biomarkers to track treatment response. Furthermore, formulation strategies such as nanoemulsions, encapsulated delivery systems, or co-administration with BBB-permeable agents may improve CNS uptake. Significantly, dietary interventions, including ketogenic diets and mixed supplements, could be explored as additional approaches to boost endogenous butyrate and ketone production.

Considering the complexity of AD pathophysiology, butyrate and lauric acid should be included in a combination therapy approach, potentially alongside current anti-amyloid drugs or anti-inflammatory agents. These synergistic strategies could enhance therapeutic outcomes while reducing side effects. Finally, large-scale, long-term studies in diverse patient groups are essential to understand individual differences and establish precise treatment guidelines.

## 10. Conclusions

AD and age-related cognitive decline are multifactorial, involving genetic, environmental, dietary, vascular, and even viral influences. Lipid-based metabolic dysfunction can exacerbate these conditions by disrupting lipid metabolism in the brain, suggesting that lipid supplementation or compounds that improve the lipid profile may be beneficial. In this context, both butyrate and lauric acid demonstrate potential. Butyrate, a SCFA produced by gut microbiota, offers neuroprotective benefits through mechanisms such as histone deacetylase inhibition, anti-inflammatory modulation, mitochondrial support, and maintenance of BBB integrity. Similarly, lauric acid, a medium-chain fatty acid derived from sources like coconut oil, may enhance brain energy metabolism and reduce neuroinflammation while promoting amyloid-β clearance. Although their exact mechanisms remain to be fully elucidated, the supplementation of these lipid-modulating compounds could help restore a healthy lipid profile and counteract neurodegenerative processes in AD. Future research should prioritize the optimization of delivery mechanisms and elucidation of molecular mechanisms to fully realize the therapeutic potential of butyrate and lauric acid.

## Figures and Tables

**Figure 1 nutrients-17-02286-f001:**
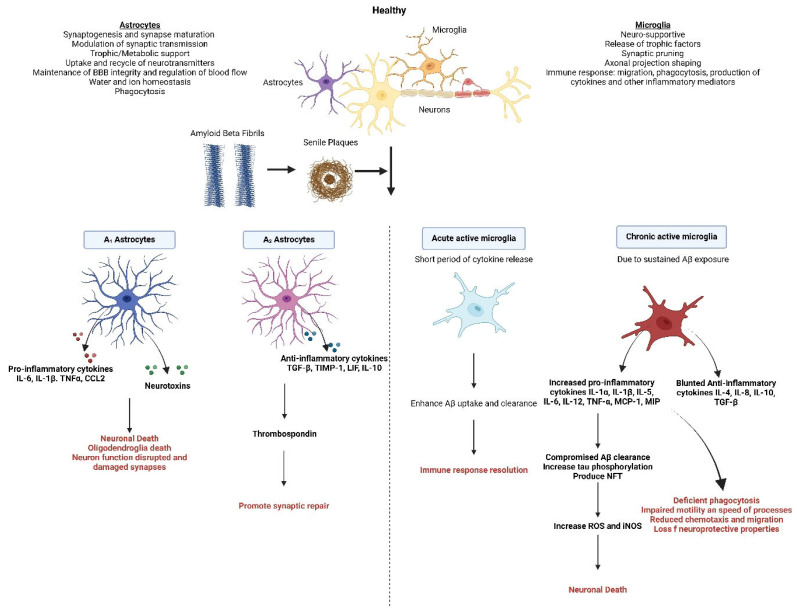
Astrocytes and microglia in Alzheimer’s disease. This figure illustrates the dual roles of astrocytes and microglia in maintaining neuronal health under normal conditions and their detrimental contributions in response to Aβ pathology. In the healthy brain, astrocytes support synaptogenesis, synapse maturation, synaptic transmission, and metabolic processes while maintaining BBB integrity, water/ion homeostasis, neurotransmitter recycling, and phagocytosis. Conversely, microglia play neuro-supportive roles, including synaptic pruning, axonal projection shaping, and immune responses such as phagocytosis and cytokine production. Upon Aβ accumulation and the formation of senile plaques, astrocytes polarize into two distinct states: A1 astrocytes, which release pro-inflammatory cytokines (e.g., IL-6, IL-1β, TNFα) and neurotoxins, leading to neuronal and oligodendroglia death, and A2 astrocytes, which secrete anti-inflammatory cytokines (e.g., TGF-β, IL-10) and thrombospondin, promoting synaptic repair. Similarly, microglia exhibit dynamic responses; acute activation involves short-term cytokine release, enhancing Aβ clearance, and resolving inflammation. However, chronic activation, driven by prolonged Aβ exposure, results in excessive release of pro-inflammatory cytokines (e.g., IL-1β, TNF-α), impaired anti-inflammatory signalling (e.g., TGF-β, IL-10), reduced Aβ clearance, tau hyperphosphorylation, and NFT formation. This chronic state increases oxidative stress, impairs microglial motility, and disrupts neuroprotective functions, ultimately causing neuronal death and synaptic damage. Together, these processes highlight the critical balance astrocytes and microglia must maintain to support neural health and the devastating consequences when this balance is disrupted in neurodegenerative conditions.

**Figure 2 nutrients-17-02286-f002:**
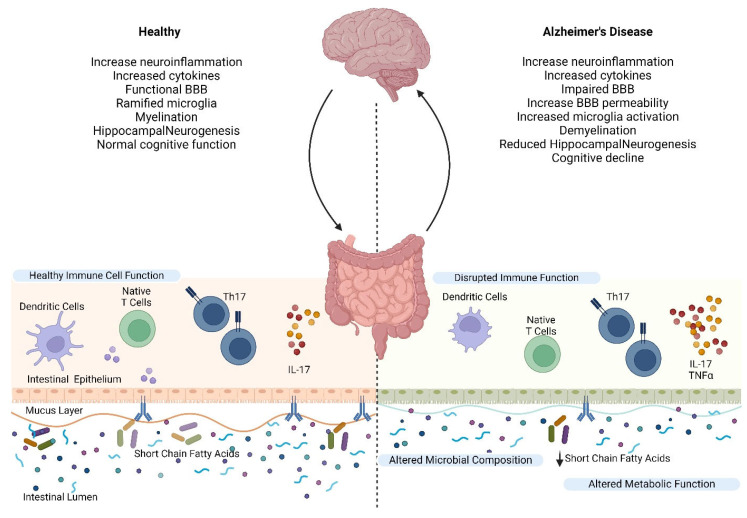
Gut–Brain Axis and the inflammatory pathway. This figure highlights the interactions between the gut and brain in both healthy conditions and AD, focusing on the role of immune and microbial functions. In a healthy state, the brain exhibits minimal neuroinflammation, functional BBB integrity, ramified microglia, efficient myelination, and hippocampal neurogenesis support normal cognitive function. The gut maintains balanced immune cell activity, including dendritic cells, native T cells, and Th17 cells, producing IL-17 within the intestinal epithelium. A robust mucus layer supports a healthy microbial composition, which produces SCFAs that sustain proper metabolic function and immune regulation. In AD, disrupted communication between the gut and brain contributes to pathology. The brain experiences heightened neuroinflammation, elevated cytokine levels, BBB impairment, increased microglial activation, demyelination, reduced hippocampal neurogenesis, and cognitive decline. In the gut, immune function is disrupted, marked by altered dendritic cell activity, dysfunctional T cells, and excessive production of pro-inflammatory mediators like IL-17 and TNFα. The mucus layer is compromised, accompanied by an altered microbial composition and reduced SCFA production, leading to impaired metabolic functions. This bidirectional gut–brain dysfunction exacerbates neuroinflammation and disease progression, underscoring the critical role of the gut–brain axis in maintaining neurological health and its disruption in Alzheimer’s disease.

## Data Availability

No new data were created.

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
