# Peer review of "The Therapeutic Potential of Butyrate and Lauric Acid in Modulating Glial and Neuronal Activity in Alzheimer’s Disease"

_nutrients, 2025, doi:10.3390/nu17142286_

Round 1

Reviewer 1 Report

Comments and Suggestions for Authors

The paper deals with a new scope and interesting aspect of AD. Neuroinflammation is important to tackle and nutritional approaches could lead to benefits as well as cut ADRs. The authors could however expand and improve the scope of the discussion by citing more relevant work.

They could also include in-depth signaling mechanisms that could be affected with this approach.

Additionally, more about the chemical nature of the agents and their main constituents would attract wider readership.

Reviewer 2 Report

Comments and Suggestions for Authors

This is an interesting and comprehensive review on the subject. The manuscript needs significant editing for language and references need to be reassessed to include citations to support all statements and to include more primary studies rather than other reviews in key areas.

Some specific points that need to be addressed:

Line 54: why is the long hyphen there? ("can shift from")

Lines 79-82: more info (i.e., what systems were these observations made in?) should be provided.

Line 98-100, sentence about MCI is missing a verb/sentence doesn't make sense as written. This is a recurring problem (i.e., sentence lines 102-107, and several other places). Manuscript needs to be read and edited more carefully for language. Some additional examples are given below, but they are not all catalogued here, the authors need to do a thorough review.

Line 118: Microglia originates should be Microglia originate

Line 148: Reproduce isn't the right word here: "Microglia are known to reproduce Aβ toxicity" --should it be induce?

Line 205: "Research has documented those aging leads to less evenly distributed microglia in the cortex" --those? Makes no sense.

Line 539-540:  Dysregulation of histone acetylation is linked to severe amyloid pathology are identified --Makes no sense

Line 554-556:  Research suggests BBB dysfunction contributes to the deposition of Aβ by activating BACE1 and γ-secretase, thus increasing the conversion of APP into Aβ and by preventing normal transport through the BBB.  --needs reference(s)

Line 294-296: "In neurological disorders characterized by BBB disruptions, astrocytes respond by tightening junctions between endothelial cells, thereby limiting the entry of immune cells into the brain" --this is an almost direct repeat of lines 250-251

Should Fig. 1 incorporate NFTs?

The figures do not appear to be referred to within the text of the article.

No references provided for lines 485-86 statement, likewise lines 492-493.

Table 1: column 3, row 4, is there missing information following 2nd "and human"?

Lines 623-624: "Existing literature shows fewer data on MCFAs and its role in human health and disease" --fewer than what?

Line 684-685: Neuropathology of AD is caused by hyperactive microglia that release NO and pro-inflammatory cytokines.  --this statement may not be completely accurate and no reference(s) are provided in support. This entire paragraph is lacking in references for most statements.

Table 2: no info under Model for row 2?

Many of the citations are of other review articles. It would be better to cite the primary papers as much as possible.

Comments on the Quality of English Language

Needs to be edited to improve comprehensibility

Reviewer 3 Report

Comments and Suggestions for Authors

This paper focuses on neuroinflammation and the role of glial cells in Alzheimer’s disease (AD), and reviews the therapeutic potential of natural compounds (butyrate and lauric acid). Alzheimer’s disease is known as a progressive neurodegenerative disorder characterized by the accumulation of amyloid beta, tau protein tangles, and extensive neuroinflammation. Elucidating its pathology and developing treatments and preventive measures are urgent tasks. Glial cells initially play a protective role (such as removing foreign substances and supplying neurotrophic factors) in the early stages of the disease, but as the disease progresses, they become chronically activated and produce inflammatory cytokines and reactive oxygen species (ROS), leading to neuronal dysfunction, loss, and cognitive decline. Current AD treatments are mainly focused on symptom relief and have limited ability to inhibit disease progression. Drugs such as Aducanumab and Lecanemab, which target amyloid beta, have also been developed, but it is undeniable that their effectiveness is limited.

Against this background, there has been increasing attention on naturally derived compounds. In this paper, we focus particularly on butyrate and lauric acid, introducing how each has demonstrated promising effects in AD models through mechanisms such as suppression of neuroinflammation, reduction of amyloid-β deposition, and promotion of neuroprotective functions. By comprehensively examining these mechanisms of action, the paper suggests the potential for new therapeutic strategies to inhibit the progression of AD, which I believe will serve as an important guideline for future medical practice.

I will comment below to further deepen the discussion.

1.The effects of butyrate and lauric acid have been demonstrated mainly in animal models and at the cellular level, and their efficacy in humans remains unverified or limited. I would like to see a deeper discussion on whether there are any plans for future measures, including clinical applications.

2.In particular, lauric acid may affect metabolism and lipid disorders at high doses, and I believe there is a lack of discussion regarding its safety and appropriate dosage. What do you think?

3.In this research report, I feel that the scientific basis and comparative examination for specifically focusing on butyrate and lauric acid among the many natural compounds are somewhat insufficient. I would like you to further elaborate on what makes these compounds more useful than other natural compounds.

4.The text deals exclusively with natural compounds and does not mention the combined effects or interactions with existing treatments. I would like a discussion on whether these compounds should be considered to have synergistic effects when used together with known drugs, or whether they might potentially have the opposite effect.

5.There are considerable individual differences in the onset and progression of AD among patients. How do you think personalized medicine and the use of biomarkers can contribute to treatment assessment, and to what extent?
